# Primate Conservation Efforts and Sustainable Development Goals in Ecuador, Combining Research, Education and Capacity Building

**DOI:** 10.3390/ani12202750

**Published:** 2022-10-13

**Authors:** Stella de la Torre, Citlalli Morelos-Juárez

**Affiliations:** 1Colegio de Ciencias Biológicas y Ambientales, Universidad San Francisco de Quito USFQ, Quito 170901, Ecuador; 2Fundación Reserva Tesoro Escondido, Quito 170901, Ecuador

**Keywords:** ecuadorian primates, environmental education, local community engagement

## Abstract

**Simple Summary:**

We present the experiences and lessons learned from two primate conservation projects in eastern and western Ecuador and analyze their contribution to the achievement of the Sustainable Development Goals (SDGs). Both projects combine research, education, and capacity building to empower and engage local communities in initiatives to protect primate habitats. The projects’ outcomes contribute to SDGs 15 (Life on land), 4 (Quality education), 2 (Zero hunger), 3 (Good health and well-being), 5 (Gender equality), 10 (Reduced inequalities), 12 (Responsible consumption and production), and 13 (Climate actions). We highlight the importance of sharing information between projects with similar scopes and the need to develop local indicators for a more objective assessment of the contribution of small-scale conservation projects to the delivery of the SDGs.

**Abstract:**

Ecuadorian primates are a diverse and ecologically important group that is facing severe conservation problems. We present the experiences and lessons learned from two primate conservation projects in eastern and western Ecuador to foster an in-depth reflection of what could be improved to enhance their contribution to the delivery of the Sustainable Development Goals (SDGs). By combining research, education, and capacity building, both projects aim to empower and engage local communities in initiatives to protect primate habitats. These efforts to enhance local environmental sustainability contribute to SDGs 15 (Life on land), 4 (Quality of education), 2 (Zero hunger), 3 (Good health and well-being), 5 (Gender equality), 10 (Reduced inequalities), 12 (Responsible consumption and production), and 13 (Climate actions). One of our findings is that community involvement in conservation activities is not always directly related to an improvement of the conservation status of primate populations. Therefore, continuous monitoring of primate populations and of other relevant indicators is key to assessing the effectiveness of the interventions. We highlight the importance of sharing information between projects with similar scopes and the need to develop local indicators for a more objective assessment of the contribution of small-scale conservation projects to the delivery of the SDGs.

## 1. Introduction

The 22 primate species that inhabit tropical and subtropical forests in Ecuador are facing severe conservation problems [1]. With an economy based on extractivism, Ecuador has one of the highest deforestation rates of the region, estimated between 56,000 and 76,000 ha per year [2,3]. Hunting, trafficking of live animals, and, possibly, zoonotic diseases are also having detrimental effects on primate populations, increasing their risk of extinction [1].

Considering their potential as flagship species, efforts to conserve Ecuadorian primates and their forests could significantly contribute to the achievement of the Sustainable Development Goals (SDGs), in particular Goal 15—Life on land, which aims to “protect, restore, and promote sustainable use of terrestrial ecosystems, sustainably manage forests, combat desertification, halt and reverse land degradation, and halt biodiversity loss” [4]. For these conservation efforts to be effective, the engagement of local communities is critical, since some of the most important drivers of deforestation in the country (namely, crop expansion, logging activities, and cattle ranching) depend on the land-use decisions of farm households [3,5]. Implementing environmental education and capacity building programs in local communities may help households to make more informed and environmentally sustainable land-use decisions, to improve their economic income and well-being. Therefore, strategies to enhance the engagement of local communities in conservation initiatives could also contribute to the achievement of Goal 4—Quality education [4] and other SDGs as well.

In this review, we present the experiences and lessons learned from two primate conservation projects in eastern and western Ecuador, the Bosques, Monos y Gente (BMG) project and the Tesoro Escondido Reserve (TER) project, led by us, to foster an in-depth reflection of what could be improved to enhance their contribution to the delivery of the SDGs. By combining research, education, and capacity building, both projects aim to empower local communities and engage them as the primary actors of conservation initiatives, reducing deforestation and other anthropogenic threats to biodiversity to eventually improve the conservation status of primate populations.

## 2. Methods

In this section, we present a brief description of the projects, an overview of the methodological approach we used to achieve our goals, and the methods to assess the projects’ outcomes.

### 2.1. BMG Project

The BMG project began in 2003, with the support of Universidad San Francisco de Quito (USFQ), University of Wisconsin–Madison, and Fundación VIHOMA. The project team is led by S.d.l.T. and includes students and researchers from USFQ and other institutions. Although this project began before the SDGs were adopted in 2015 [6], it is aligned with these goals. We aim to contribute to the conservation of Ecuadorian primates and their forests, east and west of the Andes, by improving education and building capacities in local communities to promote their involvement in sustainable productive activities and in conservation actions. In this review, we present three projects we consider best represent our work and contribution to the delivery of the SDGs: a region-wide environmental education program and two collaborative conservation initiatives with local communities in eastern—the Secoya experience—and western—Primates del Sur—Ecuador.

#### 2.1.1. Environmental Education

In 2003, while studying the behavior and ecology of pygmy marmosets, *Cebuella pygmaea* and *C. niveiventris*, in Ecuadorian Amazonia, our project team began an environmental education program for children of indigenous communities, based on didactic games about the ecology, behavior, and conservation of primates and their habitats [7]. In later years, we expanded the program to other rural and urban areas in the country. In rural schools, an environmental education journey typically lasted about 4 h per day. We spent two to three days in each school and visited schools once or twice every year. In urban schools, we spent one day per school.

To assess the impact of our environmental education program, at the beginning of each education journey we asked the children to answer a quiz (5 to 8 questions) about primate ecology, behavior, and conservation (details on the questions can be made available under reasonable request). During the education journey, we did not directly answer the questions of the quiz but included the answers in the information we gave to the children in the activities we carried out with them. Once we finished the journey, we asked the children to answer a second quiz with similar questions. We compared the number of correct and incorrect answers in the pre- and post-journey quizzes with chi-squared tests. Additionally, we qualitatively assessed the games and other didactic materials, based on the interest the children showed in using each of them.

#### 2.1.2. Secoya Experience

In 2005, we decided to concentrate our conservation work in a smaller area. We began a close collaboration with the Secoya nationality, an ethnic minority of about 600 people living in northeastern Ecuador [8,9], to develop a long-term project to conserve their culture and natural environments. We combined participatory research and education for building local capacities, reinforcing environmental values, and implementing sustainable productive activities that could improve the well-being of the Secoya people and the conservation of the native forests.

To estimate the impact of our project on the population density of primate species, we focused on the Western pygmy marmoset *Cebuella pygmaea*. From June through August each year, from 2005 through 2016, we carried out surveys along 3 km transects of gallery forests in three sites of the Secoya territory, to record the number and size of the groups of this primate species. We used Western pygmy marmoset as indicator of the conservation status of the primate community, because we had studied this species in that area since 2001 and also because the low number of recordings of the other eight primate species in the surveys we carried out [10] did not allow us to obtain reliable population estimates for most of the species across time.

To estimate the influence of the project on deforestation trends in the Secoya territory, we used land cover maps of the Global Forest Watch platform (globalforestwatch.org). We selected two areas of 3 × 3 km in each of the two Secoya settlements; we also selected a 3 × 3 km “control” site in a neighboring area of *mestizo* (settlers) farmers. In each selected area, we placed a 500 × 500 m grid on land-cover maps from 2005 (when we began our project), 2011 (when we were carrying out the project), and 2020 (after the project). We recorded the number of quadrats with at least one pixel of forest lost (identified by a specific color in the maps) and calculated the percentage of quadrats with forest lost (here on, “deforested quadrats”) from the total number of quadrats of each selected area in each of the three years.

#### 2.1.3. Primates del Sur

In 2016, we began this conservation initiative in La Libertad parish, in southwestern Ecuador, a highly disturbed area with few small remnants of Tumbesian dry forest. These forest remnants still host populations of two critically endangered primates, the Mantled howler *Alouatta palliata aequatorialis* and the Ecuadorian capuchin *Cebus aequatorialis* [11]. We combined participatory research, education, and capacity building to enhance local participation in sustainable practices, forest restoration, and biodiversity conservation.

To assess the impact of our project on primate populations, we carried out surveys in forest remnants, recording the number of groups and group sizes of the Mantled howler in 2016–17 and 2020–21.

### 2.2. TER Project

The TER Project is carried out in the lowland Ecuadorian Chocó, within the global biodiversity hotspot Tumbes–Chocó–Magdalena (Figure 1). The project is led by C.M.J. The area is home to a healthy population of the critically endangered brown-headed spider monkey *Ateles fusciceps fusciceps,* one of the 25 most endangered primates in the world [12]. It is also home to populations of *A. palliata* and *Cebus capucinus*, both listed as Critically Endangered by the Ecuadorian Red List of Mammals 2021 [11]. The Tesoro Escondido Reserve was established with the support of key members of local communities, researchers, and donors, with the long-term conservation of the brown-headed spider monkey as the primary goal. Deforestation over the past decades has left only 32% of lowland forest in the whole Ecuadorian Chocó [13]. The work of the Tesoro Escondido Reserve aims to mitigate the main threats primates and other species face in this area: loss of habitat due to timber extraction, expansion of the agricultural frontier, land use change, monocultures, hunting, and illegal pet trade. In 2016, the reserve started an integral conservation program aimed at raising awareness of the importance of the lowland Chocó and its species, involving local communities in conservation activities.

#### 2.2.1. Environmental Education 

The first stage of our permanent environmental education program (2016–2019) focused, on a weekly basis, on 11 local schools in 10 local communities around the Tesoro Escondido Reserve. Eight of these schools were multi-level (6–12 years old), with one teacher in one classroom, and three were federal schools, with 8th, 9th, and 10th grades participating in our activities. Occasional workshops were also delivered to a high school. The ethnicity in all schools was mixed, or settlers, as they are called in Ecuador. Settlers arrived from other provinces to the province Esmeraldas around 40 years ago, looking for better lands to farm. Their view of the forest was (and in many cases still is) a source of wood to sell and an area to be transformed into cropland. Moreover, the area has been dominated by industrial timber companies for over half a century. They provide most jobs and many basic services in the local communities. At the beginning of our workshops, we carried out a short survey to assess children’s knowledge on local biodiversity, with emphasis on primates. By the end of 2019, we asked children about the same topics. For some topics, such as illegally keeping wild animals as pets, we included teachers and parents in the workshops.

Our environmental education program stopped during the pandemic, from early 2019 until mid-2021, when we started the second stage of the program in a new format, called Environmental Clubs, which are carried out on a weekly basis outside the school hours in a communal space designated by the community. We also focus on the forest school methodology, taking the children to the reserve for two nights every three months. Each trip has a learning objective (i.e., primates, insects, plants), looking to awaken the children’s interest in and concern for the specific topic. We involve them in the scientific research we carry out there: they get to see how camera traps work, and they do primate monitoring, bird watching, and amphibian surveys. They are also involved in agroecology projects in their own communities and work every week with our environmental educators on different topics relevant to conservation and sustainability. Art in its different versions, particularly photography, is a regular tool that we use for students to express their concerns, their views, and their proposals for the communities and environment they want in their lives.

An oral evaluation was carried out before the start of Environmental Clubs in 2021, focused on what children did know of the area they live in and its species and their needs and wishes for activities in the club. This evaluation also helped us to assess our work before the pandemic and understand how knowledge has been internalized by children over the years. We evaluated the children again in February 2022.

In 2021, our environmental education program was part of a research project called “Hope in the Present”, by the University of Sussex, exploring the role of uncertain pedagogies such as photography and plasticine models as transformative tools in engaging children with environmental sustainability [14], which allowed our children to reflect through different methodologies about what sustainability means to them.

#### 2.2.2. Parabiologist Program

In 2013, inspired by previous experiences from other research groups in Papua New Guinea [15], Bolivia (Erika Cuellar), and Ecuador (FCAT Foundation), we began a parabiologist program. A parabiologist is a local person with no formal education, trained as a research assistant, who collaborates in research projects. These research assistants have worked alongside many researchers and students over the years.

#### 2.2.3. Agroecology

In 2018, an agroecology project was established with the objectives of improving food sovereignty and security, particularly for women. We started in the Cristobal Colon community with a one-year intense workshop (once every two weeks) aimed at women. An agroecology garden of 500 m^2^ was designed in a communal space and is maintained by members of the local community.

## 3. Results

### 3.1. BMG Project

#### 3.1.1. Environmental Education

From 2003 through 2006 we carried out environmental education journeys with 261 children from 4 to 16 years old in the schools of seven indigenous communities in the northern Ecuadorian Amazonia. From 2004 on, we included children from *mestizo* communities, both in rural and urban areas, reaching, by 2006, a total of 605 children. When comparing the answers of the children in the quizzes before and after an education journey, we found that the number of incorrect answers was significantly lower in the post-journey quizzes than in the pre-journey quizzes (Chi-squared *p*-values 0.03 to <0.0001, df = 1), pointing to a positive effect of our intervention, both in terms of the new knowledge that children gained about primate ecology and of their interest in conservation. Overall, carrying out these environmental education journeys for children has been so motivating and enriching that we always look forward to organizing these journeys whenever we have the chance. By 2020, when the pandemic started, we had reached about 1700 children in 16 locations in the coast, Andes, and Amazon, ranging from large cities such as Quito and Guayaquil to small rural settlements (Figure 2).

Despite these motivating results, we realized very early that our environmental education program was not enough for improving the conservation status of primate populations. Although children liked and learned from our presentations, we were only sporadically with them and the positive effect our program may have had, in increasing their environmental awareness and strengthening their environmental values, was eventually overcome by the actions and behaviors of their families and communities, which were not always environmentally friendly. Thus, we decided to concentrate our conservation work on a smaller human population and began a close collaboration with the Secoya nationality, to develop a long-term project to conserve their culture and natural environments.

#### 3.1.2. Secoya Experience

Our relationship with the Secoya began while we were conducting our primatological research. Indeed, we carried out one of the first journeys of environmental education in one of the Secoya schools. Based on these initial bonds, in 2005, we initiated the collaboration by carrying out a participatory ethnobotany study to rescue the traditional knowledge that the Secoya elders have about useful plants. With this information, we built the Sehuayejá Ethnobotany Trail in a forest remnant close to an area heavily impacted by the oil and palm industries [16]. This first experience allowed us to understand the importance of combining participatory research and education for building local capacities and implementing sustainable productive activities, which could improve the income and well-being of the Secoya people. Since then, this combination was the framework for all the activities of our collaborative conservation project.

For the education component, we targeted not only the children but also the teachers and other adults of the communities. To assure that environmental education was permanently taught in the five Secoya schools and the one high school, in 2007, we began a “teaching teachers” program thanks to an agreement with the regional office of the Ecuadorian Ministry of Education. We carried out participatory workshops every two months, for one continuous year, where we and the Secoya teachers collaboratively designed an innovative environmental education course plan that combined arts, science, environmental values, and ethics, for all the school grades. From 2008 through 2011, we met with the teachers every six months, so that they could share their experiences in the implementation of the course plans and discuss how to improve them.

To promote the participation of adults in the education and research initiatives, we invited them to join a training course for parabiologists that consisted of six 2-day workshops about the fundamentals of data gathering, analysis, and the publication of results, as well as the importance of research for the sustainable management and conservation of the Secoya natural environments. Once they approved the course, we hired them for research projects to assess the conservation status of representative wildlife, including primates, and to rescue their traditional knowledge about the Amazonian biodiversity. This parabiologist research team had 56 members of both sexes (23 women and 33 men), ages 17 through 72, actively involved in research from 2006 through 2013.

The results of the participatory research projects were published in three books distributed to all the Secoya people [9,16,17] and in scientific articles in Spanish and English journals and books [18,19,20,21]. These results not only evidenced the remarkable traditional knowledge of the Secoya about the Amazonian environments but also the impacts of deforestation and hunting on the diversity of mammals and large birds in their territory [19,22] and were the basis for implementing local regulations aimed to reduce the impacts of these activities.

To define the sustainable productive activities that were to be implemented, we carried out workshops with all the community. Based on previous experiences that several Secoya had on tourism, the first activity they proposed was, indeed, communitarian tourism.

In 2008, we initiated a collaborative network with several public and private organizations to develop a business plan, carry out training courses to improve the Secoya’s skills in all the components of the tourism program and build touristic infrastructure (including fully equipped, environmentally friendly lodges with solar energy and a system for treating black water, trails, an observation tower, an in-site transportation system, and interpretation centers in traditional Secoya houses). In 2010, the program of communitarian tourism was set and ready to begin. An internationally well-known tourist operator signed a collaborative agreement with the Secoya to help them with the program implementation. About two years later, however, the community decided to end this collaboration and tried other management strategies that were unsuccessful, so the tourism program came to an end.

Although the Secoya viewed communitarian tourism as the best strategy to diversify and improve their economy and to reinforce their culture, it was not their only strategy. While the communitarian tourism program was implemented, we were exploring the feasibility of the sustainable production of native fish and poultry, which could contribute to food security for the community and, eventually, be sold to external markets as elaborated products of the Secoya traditional gastronomy.

By 2010, small-scale poultry farming was carried out by eight Secoya families, following a protocol to reduce the potential conflict with native predators (e.g., carnivores such as tayras and ocelots) and to make the activity as sustainable as possible. Chickens were raised in enclosures appropriate to ensure their health and well-being. The families began to cultivate some fast-growing plants species (e.g., corn and native beans) to produce high quality local food for the chickens. This activity, however, ended soon because the productivity of the plants was low and required more time than the families were able to dedicate, so they decided to buy the food in the city markets. The poultry production was mainly used by the families, but some animals were cooked using a traditional Secoya recipe and sold in organic markets in Quito with other products of the Secoya cuisine. Although these food products were appreciated by consumers, the preparation process was complex and eventually the families stopped working on the initiative. The fish farming initiative was also stopped, mainly because of the difficulties in obtaining reproductive individuals from the wild stock of native fish species. Despite the failure of these two initiatives at that time, a positive outcome was that several Secoya of both sexes were trained in the basics of sustainable fish and poultry farming.

In the final assessment we carried out, when our direct involvement in this conservation project ended, we analyzed the changes in the number of groups and the group size of the pygmy marmoset in three areas of the Secoya territory and compared the extent of the deforested areas in the Secoya territory and in neighboring areas.

In the three Secoya sites, where we monitored pygmy marmoset groups from 2005 through 2016, the number of groups declined over time (Table 1). In two of these sites, San Pablo and Shushufindi, all groups eventually disappeared. We do not completely understand the causes of this population decline; we know that deforestation for small-scale agriculture, live capture of animals, and forest alteration due to strong winds affected some of the groups, but other factors such as zoonotic diseases could not be discarded [20,23] (S.d.l.T. pers. obs.). Although some of these factors were mitigated by our project, the efforts were not enough to reverse the negative trend.

We may have contributed, though, to the slowdown of the deforestation trend in the Secoya territory. The percentage of 500 × 500 m grids with deforested areas was lower in the Secoya territory than in the neighboring *mestizo* farms. The increase in deforested areas over the years was also lower in the Secoya territory (Figure 3).

What we believe was the most important contribution of our project, to conserve the Secoya culture and natural environments, is more difficult to quantify. It relates to the efforts we made to build local capacities to empower the Secoya community, so that they could successfully lead sustainability and conservation initiatives. Knowing that now some of the young Secoya we worked with are leading successful initiatives to rescue their traditional farms and the surrounding forest, while generating income by selling traditional Secoya food, motivates us to keep striving and learning from all these experiences to implement effective conservation strategies elsewhere.

#### 3.1.3. Primates del Sur

In the last six years, we have tried to apply what we learned from the Secoya experience to a new conservation initiative in La Libertad parish, in southwestern Ecuador. Here, the challenge is perhaps more complex, because the community needs to restore forests and create corridors to connect isolated patches, and there are fewer funding sources to support conservation initiatives of *mestizo* communities in the coast than there are for indigenous groups in the Amazon (S.d.l.T. pers. obs.).

For the education component, we implemented the “teaching teachers” program with the approval of the regional office of the Ecuadorian Ministry of Education. In 2018, we carried out trimestral workshops over two years, providing teachers didactic materials and developing a course plan for environmental education with them. One tangible result of our effort was the celebration of the “La Libertad International Monkey Day”, as part of the program of the official festivities of the parish. In this celebration, attended by most community members in 2018 and 2019, students and teachers of the four schools of the area participated in didactic and recreational activities promoting primate conservation. Since 2020, we have not been able to celebrate Monkey Day with the community because of the pandemic, but we are looking forward to resuming it soon.

Implementing the parabiologist program for the adults of the community took us more time, and we reached fewer participants (eight), partly because of funding limitations and also because the time budget of the farmers in La Libertad did not allow them to regularly attend workshops. The performance of the parabiologists during the pandemic was, however, outstanding and included primate monitoring and the reforestation of small corridors with native plants. We are now searching for new funds to support them in continuing this research.

When we began our collaboration with the local government, our mutual goal was to develop a sustainable tourism program, with primates as one of the main attractions. In 2019, with the collaboration of the Association of Tour Operators of El Oro Province, the regional government, and a local university (Universidad Técnica de Machala), we developed a business plan and best practice guidelines for primate tourism, but because of the pandemic we had to put this initiative on hold.

Not all the effects of the pandemic, though, were negative. When comparing the number of groups and the average group size of Mantled howlers in 2016–17, when we began the project, and in 2020–21, during the pandemic lockdown, we found an increase in the number of groups (from 11 to 21) and in the average group size (from 5 ± 2.92 to 8 ± 4.24, mean ± standard deviation). The reduction in human activities due to the lockdown, combined with an increase in environmental awareness, hopefully boosted by our project, may partly explain the positive results [24]. We are looking forward to reactivating all the project components in the next several months; future surveys of the Mantled howler population will allow us to better understand its dynamics and to assess the impact of our conservation efforts.

### 3.2. TER Project

#### 3.2.1. Environmental Education

In the first phase of our program, we reached around 800 students, ages 6–18. We started the workshops by situating children in the Ecuadorian Chocó, which sadly is not common knowledge in the schools. Then, we moved forward to the biodiversity that exists in the area, focusing on primates, particularly the brown-headed spider monkey and its identification, ecology, and conservation.

In the short survey we carried out at the beginning of our workshops, only 12% of all children, mainly the ones who lived very close to the forest or had forest on their farms, could correctly identify the three species of primates. Most of them confused howler monkeys and spider monkeys due to their similar color. Capuchins were identified easier. Kinkajous were also thought to be primates. About 85% of the children considered having a pet monkey as desirable.

A written evaluation at the end of 2019 showed that over 95% of the children involved in our workshops were able to identify the three species of primates living in the region, as well as other emblematic species such as jaguars, pumas, and macaws and small mammals like ocelots, tayras, and kinkajous. About 75% of students were informed about primate diets and habitat preferences. 

The campaign against keeping wild animals as pets resulted in children handing in two kinkajous, a white-collared peccary, and a baby brown-headed spider monkey, which were all taken to rescue centers.

Our environmental education program stopped during the pandemic, from early 2019 until mid-2021, when we started the second stage of the program, the Environmental Clubs. We currently work in three communities, thus far reaching around 120 children, ages 4–16. After six months with the Environmental Clubs, an evaluation was carried out in the clubs of the three communities in 2022. These children had spent time in the reserve and had discussions on more complex topics such as sustainability in their lives and wild animals as pets. They had also assisted with talks given by parabiologists in the reserve and participated in projects in the field, such as primate monitoring, seed identification, camera traps, and mist netting. The children had even discussed more complex ecology topics such as animal–plant interactions, jaguar-coexistence solutions, etc. Almost all the children who had visited the reserve (98%) correctly answered questions about the Chocó, primate species, emblematic flora and fauna, diet, and habitat requirements, as well as about sustainability and agroecology. We evaluated the change in perceptions and attitudes that the children we worked with had toward different subjects. While this is an ongoing process, being part of the Environmental Clubs as a regular weekly activity and, especially, going out into the forest and being part of participative research has changed how they see the forest and its inhabitants. For example, in terms of keeping wild animals as pets, since we started our environmental education program, the number of animals rescued from these communities has drastically diminished. We have not found any primates or kinkajous kept as pets in the last four years; in fact, children involved in the Environmental Clubs have called us and helped with the rescue of two sloths kept in a house in another town.

The Hope in the Present project by Sussex University served as an evaluation of how the activities at the Environmental Clubs have been internalized in the children’s lives and how, in towns where “modern development” is a priority, children have made the forest, its species (amongst them primates), rivers and organic gardens priorities for their future life and elements they want to conserve for their future. 

During this program we have worked on SDG 5 (Gender equality), as we involve girls in all activities, focusing on the message that boys and girls are equal. This is particularly important in rural Ecuador, where there is marked inequality between men and women regarding poverty, employment and education [25]. Our team has a strong female presence, especially in leadership positions, which, over the years, has made a positive impact in the minds of the girls who attend our workshops. This is evident in the decisions they make during the club activities and personally. Some of the oldest are choosing to go to university to study science-related subjects, others have lost the fear of speaking their opinions out loud and are demanding the same treatment as men in some activities, too. At the nearby high school, for example, out of a group of 30 graduates in 2020, four women and one man are currently studying biology, veterinary, and environmental engineering. For boys, it is also a learning process, as they are involved in cooking activities, sewing, and cleaning, traditionally viewed as women’s chores. During field trips, they learn from empowered women who are working in the forest as leaders, changing their perception of what the role of a woman is. 

#### 3.2.2. Parabiologist Program

When we started our primate monitoring at the Tesoro Escondido Reserve, we trained five local people in the methodology and hired three of them for a year to carry out primate surveys. In 2016, as the TER was established, we trained another five people, not only in primate surveys but also in camera traps and amphibian, reptile, bat, and bird, survey methodologies. The parabiologist program at the TER has trained over 30 people, permanently hiring 8 of them to carry out research in the reserve and assist students and researchers, as well as deliver environmental education workshops in the local communities to children (permanently) and adults (occasionally).

#### 3.2.3. Agroecology—Improving Food Sovereignty and Food Security

The agroecology garden, “Huerta Viva la Cristobal”, is maintained by members of the local community, applying different methodologies of agroecology and permaculture. It has become a model on organic agriculture, which had more impact during the pandemic when obtaining food from “outside” was difficult. Women retook having a garden in their own houses and learning the processes of planting and taking care of plants. Over 40 out of around 300 women are involved in agroecology in the town. Some of them make a living from their gardens and continue to teach other women in nearby towns. This project was also applied at schools before the pandemic, encouraging and supporting teachers to comply with the national TINI project by the Environmental Education Ministry (Land for Children and Young), with the aim to reconnect young people with nature by giving them a small garden plot to plant what they wanted. We trained teachers in agroecology and provided material and expertise in building a miniature agroecology space in each school. Many gardens turned afterwards into community gardens, showing the interest and compromise of the whole community regarding this project.

## 4. Discussion

Despite differences in methods and outcomes, our conservation projects share a common goal: to empower local communities as primary actors of conservation initiatives. They are also similar in the approach used to achieve this goal, by combining research, education, capacity building, and the development of sustainable productive activities in local communities. Independently, each project team decided to use this approach, developing convergent strategies and activities. Overall, we believe that our approach is effective, but we acknowledge that we would have benefited from sharing our experiences earlier. Therefore, in this review, we want to share these experiences and lessons learned with more people who are interested in carrying out conservation projects, to foster an in-depth reflection of what could be improved, to enhance their contribution to the delivery of the SDGs at the local and national levels.

The active involvement of local communities is key in our conservation initiatives, although the outcomes of the efforts for empowering local people may not always be what we initially expected and may also take more time than expected. Communities differ in their short- and long-term responses to projects’ interventions, depending on their socio-environmental characteristics. Constant evaluation, adaptability, and patience are needed to maintain the positive impacts of the education and capacity-building initiatives.

In our projects, education initiatives have always been science-based and have contributed to SDG 4′s (Quality education) targets by providing relevant learning opportunities for children and adults of both sexes in local communities. Our environmental education programs for children have been very well-received and have had positive results. Taking children to the forest in a planned and regular way, with activities that bring them closer to this environment, has an enormous impact on the children’s lives and on the *actions they* take every day toward conserving this environment in their lives. This change in attitudes is the best indicator of the success of an environmental education program.

The “teaching teachers” initiative to develop environmental education course plans for the local schools is a useful strategy to assure the continuity of the programs in the long term. In the two communities where we implemented this initiative, teachers were motivated to learn new didactic tools and used the course plans as the baseline for the new official plans requested by the Ministry of Education.

One of the lessons we learned is that to ensure that the positive effects of the education initiatives are long-lasting, it is better to concentrate the conservation efforts in few local communities, with whom we could closely interact in the long term. We hope, though, that we can also eventually collaborate, based on our experiences, in a nation-wide program of environmental education led by public institutions, such as the Ecuadorian Ministries of Environment and Education.

The combination of participatory research and education of the parabiologist programs offered the participants a different perspective, to understand and value their cultural and natural heritage. Furthermore, their interest in pursuing a university degree increased. In the BMG project, we obtained funding to support the B.S. careers, in Sustainable Development, of three young Secoya, who are currently leading the Secoya organization, an international NGO supporting Amazonian nationalities, and a regional public program for the development of local communities, respectively. Following their example, other young Secoya of both sexes are now studying undergraduate and graduate careers in a variety of areas.

Parabiologists trained at the TER have effectively become ambassadors of conservation in their own families and communities, transmitting knowledge, perceptions, and attitudes. Even parabiologists who have left for various reasons have changed their view toward the forest, nature, and the importance of conservation, applying this in their lives. Actively involving local people in research and valuing local knowledge and skills is a vital step for the conservation of primates and natural areas. This is particularly true for women, who locally have very few sources of personal and professional growth. Women who have participated in our parabiologist program have become more independent, empowered, and strong advocates of conservation, transmitting this to their children in a more active way than men. They have also become important role models for younger women, opening a path for the new generation in the area.

Our projects have contributed to the protection and local management of native forests that are among SDG 15′s targets [26]. Participating in the parabiologist program, for example, motivated the Secoya to create and enforce protected areas inside their territory, such as the Sehuayejá Ethnobotany and Wildlife Reserve. Our project may have also contributed to the reduced deforestation trend in the Secoya territory, compared to the neighboring *mestizo* farms. We acknowledge, though, that there are other factors not related to our project that should be considered to explain this difference in deforestation trends [27].

Another important lesson we learned is that protecting primate habitats by empowering local communities does not necessarily improve the conservation status of primate populations. The threats primates face are complex and may not always be mitigated by conservation projects. Systematic monitoring of primate populations and of other relevant indicators is important to assess the effectiveness of conservation interventions and the need for implementing new strategies [28]. In this review, we present some of our attempts to objectively evaluate the impact of our projects. This is one of our priorities, which we work on permanently.

The synergy between our contributions to SDGs 4 and 15 has allowed us to also contribute to other SDGs, including SDG 2 (Zero hunger), 2 (Good health and well-being), 5 (Gender equality), 10 (Reduced inequalities), 12 (Responsible consumption and production), and 13 (Climate actions). However, the fact that the SDGs’ indicators evaluate progress at the national or regional levels makes it difficult to objectively assess the contributions of small, local conservation projects such as ours. Developing a new set of indicators to assess local projects is a necessary step to improve the measurement and monitoring of their progress toward the achievement of the Sustainable Development Goals.

## 5. Conclusions

Sharing the experiences and results of conservation projects is important to guide and improve similar conservation pathways. Finding opportunities and spaces for information sharing should be a priority for project managers.

Ecuadorian primates are flagship species; to improve their conservation status, we need to conserve and restore the native ecosystems where they occur, directly contributing to SDG 15 (Life on land). For these efforts to be effective, it is critical to engage local communities in all the stages of conservation projects.

The combination of research, education, and capacity building is an effective strategy to empower and engage local communities in conservation initiatives. Efforts to enhance local environmental sustainability and conservation not only contribute to SDG 4 (Quality of education) and SDG 15 but also to SDGs 2 (Zero hunger), 3 (Good health and well-being), 5 (Gender equality), 10 (Reduced inequalities), 12 (Responsible consumption and production), and 13 (Climate actions).

Protecting forests by empowering local communities is not always directly related to an improvement of the conservation status of primate populations, given the complexity of the threats. Continuous monitoring of primate populations and of other relevant indicators is important to assess the effectiveness of interventions.

## Figures and Tables

**Figure 1 animals-12-02750-f001:**
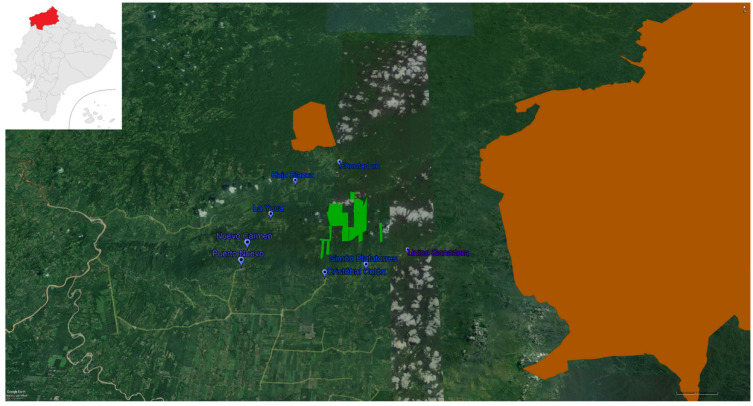
Location of the Tesoro Escondido Reserve. In light green is the Tesoro Escondido Reserve, in blue are the local communities where environmental education has taken place, and in orange are national protected areas (Pambilar Wildlife Refuge to the north and Cotacachi Cayapas National Park to the East).

**Figure 2 animals-12-02750-f002:**
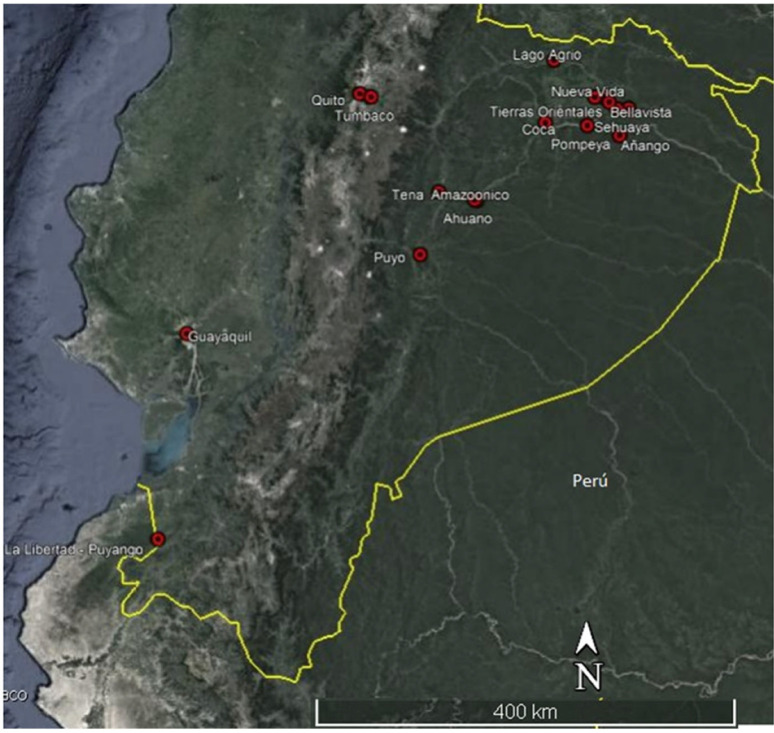
Sites where the environmental education journeys of the BMG project were carried out (2003–2020).

**Figure 3 animals-12-02750-f003:**
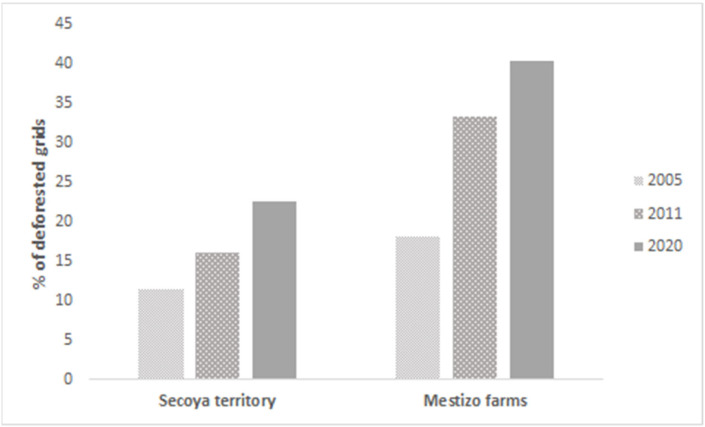
Deforestation trends, estimated by the percentage of deforested 500 × 500 m grids in 3 km^2^ areas in the Secoya territory and in nearby “mestizo” farms, in 2005, 2011, and 2020.

**Table 1 animals-12-02750-t001:** Number of groups and mean group size (±st. dev) of Western pygmy marmoset *Cebuella pygmaea* in three sites (San Pablo, Sehuaya, and Shushufindi) of the Secoya territory from 2005 to 2016.

Year	San Pablo		Sehuaya		Shushufindi	
	# of Groups	Mean Group Size (±st. dev)	# of Groups	Mean Group Size (±st. dev)	# of Groups	Mean Group Size (±st. dev)
2005	3	5.3 (±0.6)			2	5.5 (±0.7)
2006	2	4.5 (±0.7)			1	4
2007	2	5 (±1.4)			1	4
2008	2	5.5 (±2.1)	4	6.25 (±1.5)	0	
2009	2	5.5 (±0.7)	4	6.5 (±1.3)		
2010	2	5.5 (±0.7)	3	6 (±1.7)		
2011	1	6	3	7 (±1)		
2012	1	5	3	6.7 (±1.5)		
2013	1	5	3	6.3 (±0.6)		
2014	1	6	2	5.5 (±0.7)		
2015	1	5	1	7		
2016	0		1	6		

## Data Availability

The data presented in this paper are available on request from the authors.

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
