# Peer review of "Primate Conservation Efforts and Sustainable Development Goals in Ecuador, Combining Research, Education and Capacity Building"

_animals, 2022, doi:10.3390/ani12202750_

Round 1

Reviewer 1 Report

The article is interesting and clearly described and explained research on the long-term impacts of community-based conservation plans in two distinct communities. Conservation actions at both sites are evaluated in terms of the correspondence to SDGs, and (for one site) in terms of nonhuman primate and deforestation monitoring. I learned a great deal from reading the article and appreciate the opportunity to comment on it. I also recognize the complexity of sustained data collection.

I have a few minor suggestions for revision. Below, I indicate lines in the manuscript where some minor rewording might be helpful. The changed/added words are in capital letters.

Line 2: Write out SDGs in the title

Line 30: indicators ARE key…

Lines 163-164: meaning of the sentence unclear

Line 177 work, AND they do…

Line 380 Table 1: alarming decline despite such intensive and extensive efforts. Why?

3.2 TER Project: I understand that the projects entailed different methodologies and that COVID disrupted data collection at both sites, but is it possible to quantify more of the claims being made? For example, for the paragraph lines 489-498, is there information on the numbers of female students choosing university paths across time, or information on changing attitudes? Line 479 indicates that changes in perceptions were measured; is it possible to provide some of that data?

Line 464 handing IN two…

Line 468 2021 WHEN we…

Line 475 camera traps AND  mist netting

Line 479 the sentence beginning We also evaluated might be rewritten as: We evaluated the change in perceptions and attitudes that the children we worked with had toward different subjects.

Line 487 delete comma after the word life

Line 489 move the word girls to after the word involve: as we involve girls in all activities,

Line 497 sewing AND cleaning,

Line 501 add comma after the word environment

Line 507 ParabiologistS

Author Response

We thank you your valuable comments and suggestions to our manuscript "Primate conservation efforts and Sustainable Development Goals in Ecuador, combining research, education and capacity building (Animals 1917822)". Below are our point-by-point response to your comments.

Line 2: Write out SDGs in the title - We accepted the suggestion.

Line 30: indicators ARE key… - We reworded the sentence to make it clear that what is needed is the monitoring (of primate populations and of other relevant indicators).

Lines 163-164: meaning of the sentence unclear - (now Lines 190-192) We reworded the sentences to highlight the influence that timber companies have had in the communities.

Line 177 work, AND they do…(now Line 204) We accepted the suggestion.

Line 380 Table 1: alarming decline despite such intensive and extensive efforts. Why?- In the paragraph before the Table (now Lines 365-372) we explained what we know about the factors that could have caused the population decrease.

3.2 TER Project: I understand that the projects entailed different methodologies and that COVID disrupted data collection at both sites, but is it possible to quantify more of the claims being made? For example, for the paragraph lines 489-498, is there information on the numbers of female students choosing university paths across time, or information on changing attitudes? Line 479 indicates that changes in perceptions were measured; is it possible to provide some of that data? - We now present information about the number of female students choosing university careers in one high school cohort (now Lines 508-510). We also present some evidence of the changes in perceptions (now Lines 490-495).

Line 464 handing IN two… - (now Line 471) We accepted the suggestion.

Line 468 2021 WHEN we… - (now Line 475) We accepted the suggestion.

Line 475 camera traps AND  mist netting - (now Line 482) We accepted the suggestion.

Line 479 the sentence beginning We also evaluated might be rewritten as: We evaluated the change in perceptions and attitudes that the children we worked with had toward different subjects. - (now Lines 486-487) We accepted the suggestion.

Line 487 delete comma after the word life - (now Line 499) We accepted the suggestion.

Line 489 move the word girls to after the word involve: as we involve girls in all activities, - (now Lines 501-502) We accepted the suggestion.

Line 497 sewing AND cleaning, - (now Line 511) We accepted the suggestion.

Line 501 add comma after the word environment - (now Line 515) We accepted the suggestion.

Line 507 ParabiologistS - (now Line 592) We accepted the suggestion.

Reviewer 2 Report

This is an important study that evaluates the outcomes of two long-term, multi-dimensional conservation programs in Ecuador. The authors describe in detail their goals to increase awareness of primates and conservation efforts in line with UN targets, a diverse and super interesting set of approaches to involve members of the local communities in which they worked at all levels., and the outcome of primate census work at one site.  What makes this study important is the combination of long-term insights and quantitative assessments of the effectiveness of their efforts.  Their main conclusions pertain to both the positive outcomes of their efforts (e.g., students improved knowledge about primates, the parabiologist training program, local people getting involved in running sustainable agricultural and other programs) and, equally importantly, the unexpected outcomes, which included less positive impact than anticipated, the decline of the pygmy marmosets despite the program's effects, and the narrower scale at which their impact could be measured. The points that (i) conservation programs need to be carefully and critically assessed, and (ii) that these assessments need to be shared with others so that we can learn from one anothers' efforts--are extremely valuable.  The recruitment of community members to participate directly in the programs is also a highlight.

I found the paper to be very well-written, and my suggestions are simple:

If possible, please try to minimize some of the repeated sentences in the beginning, where the beginning of the summary, abstract, and introduction are currently quite similar. 

Section 3.1.2 (Primates del Sur) came as a surprise in the Results, because it had not been previewed in the methods. I wonder if there is a way to help a reader anticipate that this will come, perhaps by adding the background as a section  in the Methods so that it appears there (in Methods) in the same sequence as it does in the Results.

The authors are to be congratulated for their long-term efforts at community education and conservation. Their assessment of the effectiveness of their project sets an important standard for all such projects. I will assign this paper to my students as soon as it is published.

Author Response

We thank you your valuable comments and suggestions to our manuscript "Primate conservation efforts and Sustainable Development Goals in Ecuador, combining research, education and capacity building (Animals 1917822)". Below are our point-by-point response to your comments.

If possible, please try to minimize some of the repeated sentences in the beginning, where the beginning of the summary, abstract, and introduction are currently quite similar. - We accepted the suggestion. We have rephrased the Summary and the Introduction to reduce repeated sentences.

Section 3.1.2 (Primates del Sur) came as a surprise in the Results, because it had not been previewed in the methods. I wonder if there is a way to help a reader anticipate that this will come, perhaps by adding the background as a section in the Methods so that it appears there (in Methods) in the same sequence as it does in the Results. - We accepted the suggestion. We reorganized the subsections of the Methods to be congruent with the subsections of the Results and included a brief description and methods of the Primates del Sur initiative.